# Chronic Kidney Disease Is Associated with High Mortality Risk in Patients with Diabetes after Primary Shoulder Arthroplasty: A Nationwide Population-Based Cohort Study

**DOI:** 10.3390/diagnostics11050822

**Published:** 2021-05-01

**Authors:** Meng-Hao Lin, Su-Ju Lin, Liang-Tseng Kuo, Tien-Hsing Chen, Chi-Lung Chen, Pei-An Yu, Yao-Hung Tsai, Wei-Hsiu Hsu

**Affiliations:** 1Department of Orthopaedic Surgery, Chang Gung Memorial Hospital, Chiayi 613, Taiwan; skdan1108@gmail.com; 2Division of Nephrology, Department of Medicine, Chang Gung Memorial Hospital, Chiayi 613, Taiwan; sololin7@gmail.com; 3Division of Sports Medicine, Department of Orthopaedic Surgery, Chang Gung Memorial Hospital, Chiayi 613, Taiwan; 7572@cgmh.org.tw; 4School of Medicine, Chang Gung University, Taoyuan 333, Taiwan; skyheart0826@gmail.com; 5Division of sports medicine, Chang Gung Memorial Hospital, Chiayi 613, Taiwan; chilungchen@gmail.com (C.-L.C.); jerry03150315@gmail.com (P.-A.Y.); orma2244@cgmh.org.tw (Y.-H.T.)

**Keywords:** shoulder arthroplasty, chronic kidney disease, dialysis, diabetes mellitus, readmission, mortality

## Abstract

The number of diabetic patients with chronic kidney disease (CKD) undergoing shoulder arthroplasty is growing. This study aims to compare perioperative outcomes of shoulder arthroplasty in diabetic patients at different renal function stages. Between 1998 and 2013, a total of 4443 diabetic patients with shoulder arthroplasty were enrolled: 1174 (26%) had CKD without dialysis (CKD group), 427 (9%) underwent dialysis (dialysis group), and 3042 (68%) had no CKD (non-CKD group). Compared with the non-CKD group, the CKD (odds ratio [OR], 4.69; 95% confidence interval [CI], 2.02–10.89) and dialysis (OR, 6.71; 95% CI, 1.63–27.73) groups had a high risk of in-hospital death. The dialysis group had a high risk of infection after shoulder arthroplasty compared with the CKD (subdistribution hazard ratio [SHR], 1.69; 95% CI, 1.07–2.69) and non-CKD (SHR, 1.76; 95% CI, 1.14–2.73) groups. The dialysis group showed higher risks of all-cause readmission and mortality than the CKD and non-CKD groups after a 3-month follow-up. In conclusion, CKD was associated with worse outcomes after shoulder arthroplasty. Compared with those without CKD, CKD patients had significantly increased readmission and mortality risks but did not have an increased risk of surgical complications, including superficial infection or implant removal.

## 1. Introduction

Over the past 20 years, shoulder arthroplasty (SA) procedures have rapidly increased by approximately four- to five-fold [1]. SA includes hemiarthroplasty, anatomical total shoulder arthroplasty (ATSA), and reverse total shoulder arthroplasty (RTSA). Shoulder hemiarthroplasty and ATSA are indicated for shoulder end-stage arthritis, humeral head osteonecrosis, cuff tear arthropathy, and comminuted proximal humerus fracture. ATSA is reserved for individuals having glenoid pathology with intact rotator cuff and adequate bone stock [2,3]. Conversely, RTSA is indicated for irreparable rotator cuff tear arthropathy, revision procedures, and even comminuted proximal humerus fractures with low bone quality [4,5,6].

Although surgical techniques and implant designs in SA have improved, perioperative complications are persistent [7]. Common complications can be classified as surgical complications, including infection and loosening, and nonsurgical complications, including mortality and readmission. Patients with comorbidities such as chronic kidney disease (CKD) and diabetes mellitus (DM) have a high risk of unfavorable outcomes following joint arthroplasty [8,9,10,11,12,13]. Compared with patients without diabetes, patients with diabetes tend to have a high risk of readmission, prolonged hospitalization after SA, and even perioperative mortality and mobility [10,11,12]. CKD prevalence in patients undergoing SA is increasing [13] and is associated with increased complications after total joint procedures [8,9,13]. Moreover, dialysis-dependent patients receiving SA had increased risks of readmission, emergency department visits, and in-hospital death [8].

DM is the leading cause of CKD worldwide, and the number of patients with CKD has been increasing due to the high number of patients with DM as an underlying cause [14,15]. However, most relevant studies have not investigated DM and CKD coexistence after SA in patients with perioperative complications [8,10,12]. Most of the studies mentioned above may be limited by sample size and the lack of non-dialysis controls. Furthermore, the impact of CKD stages on the prognosis after total shoulder arthroplasty (TSA) remains unclear. Therefore, we designed this study to investigate the issues as mentioned above. This study compares SA outcomes between patients without CKD, patients with CKD but without dialysis, and dialysis patients. We hypothesize that patients with CKD have poor outcomes following SA compared to those without CKD.

## 2. Materials and Methods

### 2.1. Data Source

This was a retrospective cohort study based on the data from the Taiwan National Health Insurance Research Database (NHIRD), in which more than 23 million people (>99% of the Taiwanese population) are prospectively enrolled. Data recorded in the NHIRD were deidentified and selected based on the date of birth, gender, residential area, diagnostic codes according to the International Classification of Diseases, Ninth Revision, Clinical Modification (ICD-9-CM), medications, and surgical procedures. Further information regarding National Health Insurance (NHI) and the NHIRD has been described in previous publications [16,17,18]. This study was approved by and performed under the guidance of the Ethics Institutional Review Board of authors’ institution. Patient information was de-identified; hence, the requirement for patient consent was waived.

### 2.2. Study Population

All patients diagnosed with type 2 DM who received SA (including hemiarthroplasty, ATSA, and RTSA) between 1 January 1998, and 31 December 2013, were identified. All the patients with DM met the criteria of a minimum of five outpatient visits and prescriptions of hyperglycemia drugs [19,20]. The index date of hospitalization was defined as the date of patient admission for SA. The follow-up period was recorded from the index date of hospitalization to the date of event occurrence, death, or until 31 December 2013, whichever occurred first.

To identify relevant outcomes, patients involved in a vehicle accident, multiple traumas, and previous implant-related infections were excluded. Furthermore, we excluded patients with other conditions that could affect postoperative outcomes, including a history of malignancy, immune diseases (such as systemic lupus erythematosus, multiple sclerosis, rheumatoid arthritis, etc.), and patients receiving renal transplantation. The final cohort consisted of 4443 patients. These patients were then classified into three groups according to their renal function: patients who had no diagnosis of CKD (non-CKD group), those with CKD not undergoing dialysis (CKD group), and those receiving dialysis (dialysis group). The CKD diagnosis was confirmed based on ICD-9-CM codes; this approach has been verified in the previous studies [21,22,23]. The CKD diagnosis was based on National Kidney Foundation Kidney Disease Outcomes Quality Initiative guidelines: (1) evidence of kidney injury for ≥3 months, with or without decreased glomerular filtration rate (GFR); or (2) GFR < 60 mL/min/1.73 m^2^ ≥ 3 months, with or without kidney injury [24]. The kidney injury manifests either pathologic abnormalities or markers of kidney injury, including abnormalities in the composition of the blood (e.g., renal tubular acidosis, nephrogenic diabetes insipidus, Fanconi syndrome, etc.) or urine (e.g., albuminuria, abnormalities in the urine sediments), or abnormalities in imaging tests (e.g., polycystic kidney disease, hydronephrosis, small “echogenic” kidneys). The dialysis status was confirmed if a catastrophic illness certificate was present in the Taiwan NHIRD. The patient enrollment process is presented in Figure 1.

### 2.3. Covariates

Covariates were demographics (age, sex), socioeconomic factors (monthly income and urbanization level of the residence), cumulative hospital volume of SA, surgery duration, ten comorbidities (stroke, chronic obstructive pulmonary disease, heart failure, coronary heart disease, hyperlipidemia, cardiac dysrhythmia, myocardial infarction, hypertension, dementia, osteoporosis), and the Charlson Comorbidity Index (CCI) score. Chronic medical comorbidities were confirmed if the records indicated at least two outpatient visits or one inpatient visit in the previous year [20]. Most of these comorbidities were diagnosed based on ICD-9-CM diagnostic codes; this approach has been previously validated [20] (Appendix A). In-hospital outcomes, including new-onset venous thromboembolism, delirium, urinary tract infection, and pneumonia, were identified using the ICD-9-CM codes of outpatient diagnoses. Debridement, infection, the need for transfusion, and intensive care unit (ICU) stay were detected using Taiwan NHI reimbursement codes. Moreover, hospitalization length, in-hospital death, and medical costs were recorded.

### 2.4. Outcomes

The primary outcome was a postoperative infection, which was identified using inpatient NHI reimbursement codes. Surgery-related infection was defined as superficial or deep wound infection during the follow-up period. Superficial wound infection was defined as readmission for antibiotics treatment only. Deep wound infection was defined when readmission was required for surgical debridement with or without implant removal. The database validity is guaranteed because all medical claims must be verified by reimbursement specialists and through peer review. Secondary outcomes included all-cause readmission and mortality. Withdrawal from the NHI program indicated death [25].

### 2.5. Statistical Analysis

Patient baseline characteristics among the three study groups (non-CKD vs. CKD vs. dialysis groups) were compared using one-way analysis of variance for continuous variables and the chi-square test for categorical variables. Bonferroni multiple comparison was applied when the overall result was statistically significant. The risk of categorical in-hospital outcomes (i.e., in-hospital death) among groups was compared using logistic regression analysis. Continuous in-hospital outcomes (i.e., hospital days) among groups were compared using linear regression analysis. Regarding the time to event outcomes, the risk of all-cause mortality among groups was compared using the Cox proportional hazard model. The incidence of postoperative infection and all-cause readmission among groups was compared using the Fine and Gray subdistribution hazard model. Death during follow-up was considered a competing risk. The aforementioned regression models were adjusted for all possible confounders listed in Table 1, except that the follow-up period was replaced with the index date. Lastly, we made a subgroup analysis by splitting the study periods into the years 1998–2005 and 2006–2013 to evaluate the potential impact of prosthesis and surgical technique on the association between renal status and risks for outcomes of interest. Two-tailed *p*-value of <0.05 was considered statistically significant, and no adjustment of multiple testing (multiplicity) was made. All statistical analyses were performed using SAS version 9.4 (SAS Institute, Cary, NC, USA). The adjusted cumulative incidence function and adjusted survival rate were generated using the “%dacif” and “%adjsurv” SAS macros [26,27].

## 3. Results

### 3.1. Patient Characteristics

A total of 4443 patients with diabetes who underwent SA were analyzed in the study. Among these patients, 1174 (26%) had non-dialysis CKD, 227 (5%) underwent dialysis, and the remaining 3042 (68%) had no CKD (Figure 1). During the study period, the proportion of individuals with non-dialysis CKD and dialysis increased (*p*-trend < 0.001) (Figure 2 and Appendix A).

The dialysis group was the youngest, followed by the non-CKD and CKD groups. The dialysis group had the highest CCI score and prevalence of comorbidities, including stroke, heart failure, coronary heart disease, myocardial infarction, hypertension, and osteoporosis, followed by the CKD and non-CKD groups (Table 1).

### 3.2. In-Hospital Outcomes

Regarding in-hospital outcomes, compared with the non-CKD group, the CKD group had a high risk of blood transfusion, ICU stay, and in-hospital death after SA. Moreover, the dialysis group had a high risk of blood transfusion and in-hospital death, extended hospital stays, and high medical costs. The dialysis group had a higher risk of blood transfusion than the CKD group (Table 2).

### 3.3. Late Outcomes

Table 3 shows the results of postoperative events. After SA, the dialysis group had a higher risk of any infection than the CKD (subdistribution hazard ratio (SHR), 1.69; 95% confidence interval [CI], 1.07–2.69) and non-CKD (SHR, 1.76; 95% CI, 1.14–2.73) groups. Noticeably, no significant difference was observed in the risk of any infection between the CKD and non-CKD groups (Figure 3a). In the early stage (at 30 days), the CKD group had a higher risk of all-cause readmission than the non-CKD group. After that (at 90 days, one year, and the end of follow-up), the dialysis group showed a higher risk of all-cause readmission than the other two groups (Figure 3b). The result of all-cause mortality was similar to that of all-cause readmission (Figure 3c). Furthermore, subgroup analysis failed to find any difference on outcomes between two different time periods (Appendix A).

## 4. Discussion

This study’s principal findings indicated that patients with CKD had a higher risk of in-hospital complications, including transfusion, ICU stays, and even in-hospital death. During follow-up, CKD increased the risks of infection, readmission, and mortality following SA. Dialysis patients have higher risks of infection, readmission, and mortality than those who do not receive dialysis. Therefore, caution is required during perioperative care in this patient group.

DM and CKD are independent risk factors for perioperative complications following SA and joint replacement surgery [8,10,11,12,28,29]. Cancienne et al. [8] reported that dialysis is a significant independent risk factor for complications after SA. Furthermore, hemodialysis involves an even higher risk of postoperative complications than peritoneal dialysis [8]. In the current study, the proportion of patients with impaired renal function undergoing SA increased in the 15 years in question. The risks of in-hospital and postoperative complications were higher after SA in patients with CKD than in those without CKD.

Patients with CKD have multisystem insufficiency and imbalanced electrolytes, which may jeopardize their immune function and make them susceptible to infection [30]. In lower extremity arthroplasty, patients with DM and CKD showed surgical-site infection, which is also a leading cause of 30-day readmission following primary total knee or total hip arthroplasty [28,29]. However, patients with CKD who received hip hemiarthroplasty did not increase infection risk [22,31]. Previous studies on SA have shown that the infection rate was approximately 1%, and female sex and older age were reported to be protective factors for prosthetic joint infection [32,33]. In the current study, the deep infection rate was approximately 4% after SA. After adjusting for potential confounders, dialysis patients had a higher risk of any infection–except deep infection–after SA, than non-dialysis patients.

Repeated hospitalization is common in patients with CKD during postoperative care, increasing the mortality risk and health care burden [34]. Both patients with CKD and end-stage renal disease were frequently admitted because of infection, cardiovascular problems, and deteriorated renal function, such as electrolyte imbalance, uremic syndromes, and fluid overload [35]. Similar to the findings of other studies on lower extremity arthroplasty [21,36], patients with CKD did not have an increased risk of worse surgery-related complications after SA, including infection or revision, than patients without CKD. Thus, the main reasons for readmission of patients with CKD after SA were medical comorbidities.

CKD patients were associated with an increased risk of mortality after arthroplasty, both at the index date of surgery and during follow-up. Patients with CKD who had undergone lower extremity arthroplasty had a significantly higher risk of mortality than patients without CKD [22,28,36,37,38,39]. A database study showed that the combination of CKD and diabetes had a synergistic effect on the mortality risk, compared with CKD or diabetes alone during the first postoperative year after lower extremity arthroplasty [37]. However, in the current study, the mortality rate was less than that reported in previous studies on lower extremity arthroplasty [22,36,39]. This may be because, first, the nature of the anatomical location in lower extremity procedures may have more significant effects on mobilization and daily activities than SA. Second, patients with malignancy, multiple trauma, vehicle accident trauma, immune disease, and previous implant infections were excluded. For inpatient mortality, dialysis-dependent patients had approximately ten times more risk than non–dialysis-dependent patients, and the inpatient mortality rate was approximately 0–14% in elective total hip arthroplasty or total knee arthroplasty [38,39]. Similar to a recent study [13], the present study found that non-dialysis and dialysis-dependent patients with CKD had a five- to seven-fold higher risk of in-hospital death than those with preserved renal function. This may be because of the strong association between deteriorated renal functions and cardiovascular risk [40,41]. Moreover, various organ diseases, including liver disease, congestive heart failure, and cardiac arrhythmia, are associated with mortality in dialysis-dependent patients [39].

To the best of our knowledge, this is the first study to assess the effects of renal function on SA outcomes in patients with diabetes. This study’s strengths include the use of a large-size nationwide population treated at different institutions and thorough follow-up. With the advantage of the high coverage of NHI, this retrospective study included all of the patients with diabetes in Taiwan, irrespective of where they stayed (home or nursing house), their resident area (urban or country), and visiting hospital level (regional or medical center). This reinforces the general applicability of the study results. 

Nevertheless, this study has several limitations, similar to other database studies. First, we cannot control confounding variables that may not be available in the database. Furthermore, misclassification bias should be mentioned in the database study. Owing to the lack of laboratory data and detailed medical records from the NHIRD, the misclassification of patients with diabetes or CKD is inevitable. Moreover, this prevents further quantitative analysis of renal function. However, to minimize the bias, we used the condition of a minimum of five diabetes-related outpatient visits and prescriptions of hyperglycemia drugs to confirm type 2 diabetes [19,20]. Regarding dialysis-dependent patients, rigorous regulations have been established to verify the dialysis status in the NHIRD [42]. Second, some confounding bias was present due to the lack of detailed medical records, including details for surgery, time to surgery, radiography, and even body mass index. To minimize the impact of confounding bias, we adjusted these parameters when analyzing the outcomes among groups. Thus, the difference between the three groups may be minimized after considering the impact of bias. Third, we could not evaluate the quality of prosthesis implantation due to the lack of radiologic information. Fourth, TSA and RTSA share the same procedure code, and SA was not classified based on design or cement usage. Additional comparative studies focusing on the issues mentioned above are warranted. Fourth, frailty is known to be related to prognosis, including readmission and mortality. However, we were not able to include frailty as a covariate because there were no ICD-9 diagnostic codes for frailty. Even so, we included several comorbidities related to frailty such as stroke, heart failure, myocardial infarction, dementia, and CCIs, which represented the overall health of the included patients. We adjusted all mentioned confounders in the regression models of the outcomes, which might decrease the impact of the frailty on the findings.

## 5. Conclusions

In this nationwide, population-based cohort study, we analyzed the outcomes of SA in patients with diabetes who had different renal function statuses. We found that CKD was associated with worse outcomes after SA. Patients with CKD had significantly high readmission and mortality risks but did not exhibit an increased risk of surgical complications, including superficial infection or implant removal, compared with patients without CKD.

## Figures and Tables

**Figure 1 diagnostics-11-00822-f001:**
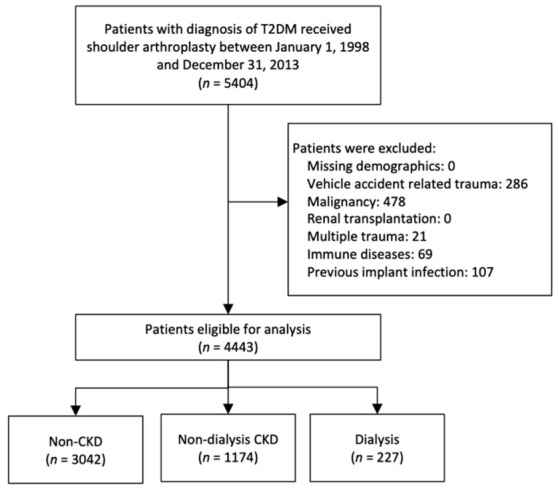
Flowchart for patient inclusion.

**Figure 2 diagnostics-11-00822-f002:**
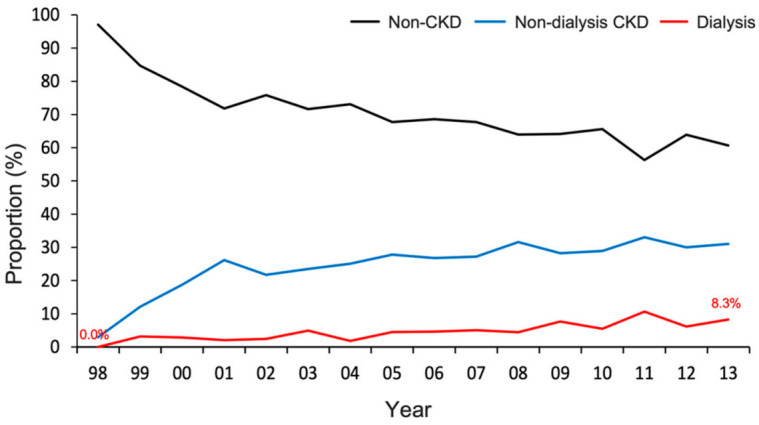
Distribution of renal function status for patients with diabetes who received shoulder arthroplasty from 1998 to 2013. The proportion of patients with non-dialysis CKD and those requiring dialysis stably increased across the study period (*p*-trend < 0.001). CKD, chronic kidney disease.

**Figure 3 diagnostics-11-00822-f003:**
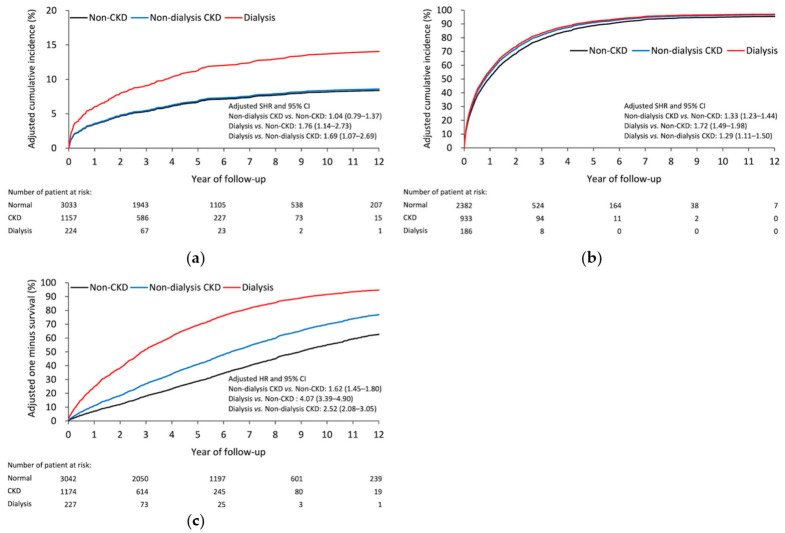
Adjusted cumulative incidence of (**a**) postoperative infection and (**b**) all-cause readmission. (**c**) Adjusted survival rate of all-cause mortality in patients with diabetes having different renal function statuses. CKD, chronic kidney disease.

**Table 1 diagnostics-11-00822-t001:** Baseline characteristics of patients with diabetes who received shoulder arthroplasty according to renal function status.

Variable	Non-CKD (*n* = 3042)	Non-Dialysis CKD (*n* = 1174)	Dialysis (*n* = 227)	*p*-Value
Age (years)	69.6 ± 11.9	71.8 ± 10.6 ^a^	65.4 ± 10.3 ^a,b^	<0.001
Age group				
<60 yrs.	682 (22.4)	173 (14.7) ^a^	83 (36.6) ^a,b^	<0.001
60–80 yrs.	1766 (58.1)	745 (63.5) ^a^	125 (55.1)	
>80 yrs.	594 (19.5)	256 (21.8)	19 (8.4) ^b^	
Female sex	2112 (69.4)	831 (70.8)	144 (63.4)	0.088
Hospital level				
Medical center	1134 (37.3)	411 (35.0)	91 (40.1)	0.041
Region hospital	1268 (41.7)	537 (45.7)	102 (44.9)	
District hospital or clinics	640 (21.0)	226 (19.3)	34 (15.0)	
Hospital volume (surgeries)				0.013
Q1 (1–130)	764 (25.1)	287 (24.4)	46 (20.3)	
Q2 (132–262)	772 (25.4)	278 (23.7)	58 (25.6)	
Q3 (281–544)	762 (25.0)	313 (26.7)	81 (35.7) ^b^	
Q4 (598–1427)	744 (24.5)	296 (25.2)	42 (18.5)	
Monthly income, USD				0.130
≤$596	1095 (36.0)	414 (35.3)	70 (30.8)	
$597–$760	1236 (40.6)	510 (43.4)	94 (41.4)	
>$760	711 (23.4)	250 (21.3)	63 (27.8)	
Urbanization level				0.401
Low	437 (14.4)	192 (16.4)	33 (14.5)	
Moderate	1002 (32.9)	366 (31.2)	77 (33.9)	
High	917 (30.1)	333 (28.4)	60 (26.4)	
Very high	686 (22.6)	283 (24.1)	57 (25.1)	
Duration of surgery				0.133
≤4 h	880 (28.9)	362 (30.8)	77 (33.9)	
>4 h	2162 (71.1)	812 (69.2)	150 (66.1)	
Comorbidity				
Stroke	463 (15.2)	253 (21.6) ^a^	56 (24.7) ^a^	<0.001
COPD	264 (8.7)	158 (13.5) ^a^	18 (7.9)	<0.001
Heart failure	534 (17.6)	323 (27.5) ^a^	85 (37.4) ^a,b^	<0.001
Coronary heart disease	556 (18.3)	380 (32.4) ^a^	76 (33.5) ^a^	<0.001
Hyperlipidemia	706 (23.2)	353 (30.1) ^a^	46 (20.3) ^b^	<0.001
Cardiac dysrhythmia	212 (7.0)	113 (9.6) ^a^	22 (9.7)	0.009
Myocardial infarction	175 (5.8)	117 (10.0) ^a^	30 (13.2) ^a^	<0.001
Hypertension	2026 (66.6)	933 (79.5) ^a^	187 (82.4) ^a^	<0.001
Dementia	221 (7.3)	113 (9.6) ^a^	11 (4.8)	0.009
Osteoporosis	1426 (46.9)	577 (49.1)	142 (62.6) ^b^	<0.001
CCI score	2.0 ± 2.0	3.0 ± 2.0 ^a^	5.0 ± 2.0 ^a,b^	<0.001
Follow up duration (years)	5.5 ± 3.9	3.8 ± 3.0 ^a^	2.7 ± 2.3 ^a,b^	<0.001

Data are expressed as mean ± standard deviation or frequency (percentage). “a” and “b” indicates significantly different from “non-CKD” and “CKD” groups, respectively, in the Bonferroni multiple comparisons. Abbreviations: CKD, chronic kidney disease; STD, standardized difference; Q, quartile; USD, United States Dollar; COPD, chronic obstructive pulmonary disease; CCI, Charlson comorbidity index.

**Table 2 diagnostics-11-00822-t002:** In-hospital outcomes of diabetic patients with shoulder arthroplasty according to renal function status.

	Number of Events (%)			Adjusted OR or β ^‡^					
Outcome	Non-CKD (*n* = 3042)	Non-Dialysis CKD (*n* = 1174)	Dialysis (*n* = 227)	Non-Dialysis CKD vs. Non-CKD		Dialysis vs. Non-CKD		Dialysis vs. Non-Dialysis CKD	
				OR or β (95% CI)	*p*-Value	OR or β (95% CI)	*p*-Value	OR or β (95% CI)	*p*-Value
Categorical									
Newly-onset VTE	7 (0.23)	4 (0.34)	0 (0)	1.68 (0.46–6.20)	0.436	NA	NA	NA	NA
Delirium	5 (0.16)	4 (0.34)	0 (0)	1.76 (0.40–7.78)	0.455	NA	NA	NA	NA
Debridement	44 (1.45)	16 (1.36)	0 (0)	1.02 (0.56–1.86)	0.949	NA	NA	NA	NA
Infection	76 (2.5)	23 (2.0)	2 (0.88)	0.87 (0.53–1.42)	0.573	0.39 (0.09–1.63)	0.196	0.45 (0.10–1.96)	0.284
UTI	106 (3.5)	56 (4.8)	5 (2.2)	1.16 (0.82–1.64)	0.391	0.58 (0.23–1.49)	0.258	0.50 (0.20–1.29)	0.153
Pneumonia	50 (1.6)	28 (2.4)	2 (0.88)	1.19 (0.73–1.94)	0.492	0.66 (0.15–2.82)	0.569	0.55 (0.13–2.40)	0.428
Transfusion	1922 (63.2)	897 (76.4)	186 (81.9)	1.87 (1.59–2.21)	<0.001	2.98 (2.07–4.30)	<0.001	1.59 (1.09–2.33)	0.016
ICU stay	106 (3.5)	83 (7.1)	15 (6.6)	1.68 (1.23–2.29)	0.001	1.79 (0.98–3.26)	0.058	1.07 (0.58–1.95)	0.833
In-hospital death	9 (0.29)	17 (1.4)	3 (1.3)	4.69 (2.02–10.89)	<0.001	6.71 (1.63–27.73)	0.009	1.43 (0.38–5.35)	0.595
Continuous									
Hospital days	10.0 ± 9.0	12.0 ± 11.0	12.0 ± 10.0	1.79 (1.13–2.45)	<0.001	2.29 (0.96–3.61)	0.001	0.50 (−0.89–1.88)	0.481
Cost (USD × 10^3^)	3.6 ± 2.4	4.2 ± 4.0	4.5 ± 2.4	0.47 (0.27, 0.67)	<0.001	0.85 (0.45, 1.26)	<0.001	0.38 (−0.04, 0.80)	0.077

^‡^ The model was adjusted for all covariates listed in Table 1 in which the follow-up year was replaced with the index date. Abbreviations: CKD, chronic kidney disease; OR, odds ratio; β, regression coefficient; CI, confidence interval; VTE, venous thromboembolism; NA, not applicable; UTI, urinary tract infection; ICU, intensive care unit; USD, United States Dollar. Data are expressed as frequency (percentage) or mean ± standard deviation.

**Table 3 diagnostics-11-00822-t003:** Late outcomes of diabetic patients with shoulder arthroplasty according to renal function status.

	Number of Events (%)			Adjusted Hazard Ratio and 95% CI ^‡^					
Outcome	Non-CKD (*n* = 3042)	Non-Dialysis CKD (*n* = 1174)	Dialysis (*n* = 227)	Non-Dialysis CKD vs. Non-CKD		Dialysis vs. Non-CKD		Dialysis vs. Non-Dialysis CKD	
				HR or SHR (95% CI)	*p*-Value	HR or SHR (95% CI)	*p*-Value	HR or SHR (95% CI)	*p*-Value
Infection									
Superficial infection	169 (5.6)	61 (5.3)	18 (8.0)	1.14 (0.83–1.56)	0.416	1.63 (0.97–2.75)	0.067	1.43 (0.83–2.47)	0.197
Debridement	59 (1.9)	16 (1.4)	6 (2.7)	0.94 (0.52–1.70)	0.842	1.43 (0.59–3.50)	0.431	1.52 (0.58–4.02)	0.398
Implant removal	72 (2.4)	23 (2.0)	6 (2.7)	1.09 (0.67–1.75)	0.738	1.34 (0.58–3.10)	0.501	1.23 (0.51–2.96)	0.642
Any infection	228 (7.5)	75 (6.5)	26 (11.6)	1.04 (0.79–1.37)	0.780	1.76 (1.14–2.73)	0.011	1.69 (1.07–2.69)	0.025
All-cause readmission									
At 30 days	293 (9.7)	170 (14.7)	30 (13.4)	1.39 (1.15–1.70)	0.001	1.28 (0.85–1.92)	0.234	0.92 (0.61–1.38)	0.679
At 90 days	565 (18.6)	306 (26.4)	76 (33.9)	1.33 (1.15–1.54)	<0.001	1.62 (1.27–2.09)	<0.001	1.22 (0.95–1.58)	0.125
At 1 year	1138 (37.5)	609 (52.6)	141 (62.9)	1.42 (1.29–1.58)	<0.001	1.74 (1.46–2.08)	<0.001	1.22 (1.02–1.47)	0.031
At the end	2382 (78.5)	933 (80.6)	186 (83.0)	1.33 (1.23–1.44)	<0.001	1.72 (1.49–1.98)	<0.001	1.29 (1.11–1.50)	0.001
All-cause mortality									
At 30 days	24 (0.79)	33 (2.8)	7 (3.1)	3.33 (1.93–5.75)	<0.001	4.82 (1.95–11.89)	0.001	1.45 (0.61–3.43)	0.402
At 90 days	62 (2.0)	58 (4.9)	18 (7.9)	2.22 (1.53–3.21)	<0.001	4.74 (2.69–8.36)	<0.001	2.14 (1.22–3.76)	0.008
At 1 year	199 (6.5)	148 (12.6)	43 (18.9)	1.83 (1.47–2.28)	<0.001	3.56 (2.50–5.06)	<0.001	1.95 (1.36–2.78)	0.000
At the end	1191 (39.2)	555 (47.3)	143 (63.0)	1.62 (1.45–1.80)	<0.001	4.07 (3.39–4.90)	<0.001	2.52 (2.08–3.05)	<0.001

^‡^ The model was adjusted for all covariates listed in Table 1 in which the follow-up year was replaced with the index date. Abbreviations: CKD, chronic kidney disease; HR, hazard ratio; SHR, subdistribution hazard ratio; CI, confidence interval. Data were given as frequency (percentage).

## Data Availability

The data are not publicly available due to the regulations of Taiwan welfare department for NHIRD.

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
