# Peer review of "Chronic Kidney Disease Is Associated with High Mortality Risk in Patients with Diabetes after Primary Shoulder Arthroplasty: A Nationwide Population-Based Cohort Study"

_diagnostics, 2021, doi:10.3390/diagnostics11050822_

Round 1
Reviewer 1 Report
The study by Lin et al from Taiwan reports the complications of shoulder arthroplasty in patients with diabetes with or without kidney disease. It's a large retrospective study which was statistically well performed and the paper is well written. The results of the study are of interest but not surprising and not novel. Based on this study it is not possible to change the current clinical practice.
Major:
- I have the impression that these results are driven by frailty of the patients. Can the authors comment on that? Please include indicators of frailty into the study, if possible.
- line 89: what is meant with immune disease? Patients with autoimmune diseases such as rheumatoid arthritis have risk at shoulder joint problems but also diabetes (due to steroids) and renal complications (sometimes directly because of RA). I think the authors should specifiy what this group entails and why they decided to exclude these patients. Also considering their statement in the discussion (line 28) about the immune status.
Minor:
Line 56: DM is a leading cause of diabetic nephropathy in developed countries but also in developing countries.
Author Response
We made responses to all your comments. Please see the attachments

Reviewer 2 Report
In my opinion the presented paper has some weak points, which should be clarified before further processing.
- I am not convinced by classifying patients into groups based on ICD. It is easy to classify patients with diabetes only and those receiving renal replacement therapy, so in end-stage renal disease but how do you recognise those with chronic kidney disease.
- Non-CKD group "patients with normal renal function" What do you mean by normal renal function?
- It should be bear in mind, that according to KDIGO guidelines published in 2012, chronic kidney disease (CKD) might be diagnosed even in presence of serum creatinine concentration within the normal range.
- "Evidence of kidney injury in blood or urine test" What do you mean by that?
- Criteria used for classification patients into group "non-CKD" and "CKD" are not clear and might be misleading.
Author Response
We made responses to all your comments. Please see the attachment.

Round 2
Reviewer 2 Report
Thank you for the response you have made. I totally accept it.